# PCA Feature Alignment is Sufficient for Building Graph Foundation models

## Abstract

Graph foundation models (GFMs) aim to pretrain graph neural networks (GNNs) that can generalize to new graph datasets in a zero-shot manner, requiring little or no additional training. This goal is challenging because graph data collected from diverse domains often exhibit significantly different node features and topological structures, and standard GNNs are sensitive to such variations. Prior efforts either restrict learning to a single domain (e.g., molecular graphs) or rely on heavy auxiliary pipelines that transform raw features into surrogates using large language models (LLMs) or feature-graph constructions. However, these approaches are often computationally expensive or restricted to specific domains.

In this work, we find that principal component analysis (PCA) is a simple, efficient feature alignment for GFMs. We show that PCA-aligned features satisfy two properties central to zero-shot GFM generalization: (i) *equivalence on identical datasets* (identical datasets with only feature dimensions permuted or node order permuted always generalize to invariant or equivalent graph representations) and (ii) *representation bounded on latently same datasets* (non-identical datasets with same latent space always generate graph representations within bounded distance and prediction error). Building on this alignment method, we develop a Mini-GFM framework that is trained once across multiple datasets; at generalizing time, it only requires PCA alignment of the new dataset and optimization of a shallow, task-specific linear head. Across diverse node- and graph-classification benchmarks, this approach delivers competitive zero-shot performance compared with other baselines while using substantially lower preprocessing cost. These theoretical and empirical results validate the sufficiency of PCA alignment.

## 1 Introduction

Foundation models Brown et al. (2020); Bommasani (2021) with strong zero-shot generalization have drawn increasing attention across machine learning, including graph learning. Unlike images or natural language, whose inputs lie in relatively canonical spaces (e.g., pixels with spatial locality; tokens with positional order) Mao et al. (2024), node attributes on graphs are typically (*i*) human-defined with heterogeneous scales and semantics, and (*ii*) stored in arbitrary dimension orders, even when mined from the same source. These inconsistencies make it difficult to build graph foundation models (GFMs) that generalize graph representation in a *zero-shot*[1] manner across datasets.

**Current Approaches.** Existing approaches to this challenge fall into two families. **(1) Domain-specific GNNs** Shoghi et al. (2024); Sypetkowski et al. (2024); Wang et al. (2024); Klaser et al. (2024); Jia et al. (2025); Li et al. (2025) focus on single domains (e.g., molecular graphs), where shared feature spaces simplify cross-task generalization, but limit applicability to new domains. **(2) Text-attributed graph pipelines** Liu et al. (2024); Xia et al. (2024); Hou et al. (2024); Huang et al. (2023); Kong et al. (2025); Tang et al. (2024); He et al. (2025) convert raw node features into textual descriptions, enabling embedding via large language models (LLMs). This allows broader generalization but comes at the cost of heavy preprocessing and LLM dependency. A third, emerging direction seeks **domain-agnostic surrogate features** Zhao et al. (2024); Shen et al. (2024) derived directly from node attributes to bridge datasets without additional metadata.

---

[1] In this work, "zero-shot" is referred specifically to generalizing graph presentations regardless of downstream tasks. More details are provided in Section 3.

(a) Mining the same source graph may result in different features.      (b) Existing Solutions

Figure 1: (a) Given the *same* data source, different graph miners may present the graph data in different forms, making GNN models hard to generalize. (b) Existing solutions include STAGE Shen et al. (2024), which builds a hypergraph graph for every pair of connected nodes and generates hypergraph embeddings as edge features, and GraphAny Zhao et al. (2024), which classifies given nodes by precalculated LinearGNNs first and then applies a learnable attention module to aggregate.

Recent studies by Shen et al. (2024); Zhao et al. (2024) formalized *feature permutation invariance* as a basic criterion for GFMs: if a dataset $\mathcal{D}$ is transformed into $\mathcal{D}'$ by permuting feature dimensions, a model trained on $\mathcal{D}$ should maintain comparable performance on $\mathcal{D}'$ since they are identical datasets. This is practically motivated—real-world graph datasets are rarely curated with consistent feature orders or scales (Figure 1a)—and non-trivial, as most GNNs (e.g., GCN Kipf & Welling (2017)) rely on learned projections that are sensitive to feature permutations. To address this, Shen et al. (2024) proposed **STAGE**, which constructs a surrogate *feature hypergraph* by pairing attributes of connected nodes and then uses a GNN on that surrogate to derive permutation-insensitive embeddings (Figure1b). While conceptually effective, STAGE performs embedding steps for every edge's hypergraph, which increases preprocessing cost—especially for high-dim. features and dense graphs. In **GraphAny** Zhao et al. (2024), surrogate features are generated by several statistical LinearGNNs' classification, which are aggregated towards the final result via learnable attention. It holds a fair complexity and good empirical results, while its dependence on node labels limits its applicability.

**Our Proposal.** In this paper, we find that **Principal Component Analysis (PCA)** is already a simple yet effective universal feature alignment method for GFMs. PCA naturally satisfies two critical invariance properties for zero-shot generalization on identical datasets: (i) *dimension permutation invariance*—permuting feature dimensions before PCA yields the same reduced representation, and (ii) *node permutation equivalence*—permuting node order yields the correspondingly permuted reduced features. We summarize these properties as *"identical equivalence"*.

Motivated by this insight, we propose a **Mini-GFM** training framework in which a core GNN is pretrained across datasets using PCA-aligned features. At generalization time, the target dataset is aligned via PCA, and only a lightweight task-specific downstream layer is trained. Notably, this method is simple, computationally efficient, and effective; it requires no LLMs, no handcrafted descriptors, and no surrogate feature graphs. More importantly, we make further theoretical analysis on our Mini-GFM and show that even facing non-identical datasets, our method could still hold bounded generalization errors if they share a similar latent feature space. We summarize such properties as *"representation bounded on the same latent space"*. This finding offers a deeper insight into how PCA alignment helps GFM capture and learn invariants shared in non-identical graph datasets and further generalize expressive representations on new graph data.

Our contributions are as follows.

1. **Insights on PCA's Properties** We adapt PCA as a feature-alignment mechanism for GFMs and prove that it satisfies zero-shot properties central to identical datasets generalization, including dimension permutation equivalence and node permutation equivalence.

2. **Lightweight Framework** We develop a GFM training framework where the core GNN is pretrained once and reused across datasets; zero-shot generalization requires only PCA on the target dataset and fitting a downstream linear layer. This framework is further proved to be theoretically effective on non-identical but latently the same datasets with bounded generalization error.

3. **Empirical Effectiveness** We evaluate on multiple node- and graph-classification benchmarks from diverse domains and scales, and compare with existing baselines and ablation settings, showing that PCA alignment enables effective zero-shot graph representations.

## 2 BACKGROUNDS AND PRELIMINARIES

**GNN Embedding** Let a graph be $G = \{\mathcal{V}, \mathcal{E}, \mathbf{X}\}$, which consists of the nodes $\mathcal{V}$, node features $\mathbf{X}$, and edges $\mathcal{E}$. We denote $u \in \mathcal{V}$ as a node, $e = (u, v) \in \mathcal{E}$ as an edge, and $\mathbf{X}_{(u)}$ as node $u$'s feature.

Typically, GNN learns node embeddings for graph data in two steps: (1) aggregating node features along the neighborhood and (2) updating by linear weights on the aggregated hidden. This so-called message-passing mechanism not only captures the structural information of the graph but is also sensitive to the input initial features. While different GNNs use different aggregate and combine operations, in this paper, we utilize the classic Graph Convolutional Network (GCN) Kipf & Welling (2017) as our theoretical model. The updating of GCN could be normalized as follows:

$$\mathbf{h}^l = \sigma(\mathbf{A}\mathbf{h}^{l-1}\mathbf{W}^l) \tag{1}$$

where $\mathbf{W}^l$ is the linear weight of $l$-th layer and $\mathbf{h}^l$ is the updated hidden. Initial hidden $\mathbf{h}^0 = \mathbf{X}$ and $\mathbf{A}$ is the normalized adjacency matrix representing $\mathcal{E}$. $\sigma$ is a globally used activation function. After forwarding all layers, the hidden $\mathbf{h}^L$ of the final layer $L$ is used as the node embedding. Depending on different downstream tasks, an additional linear layer with weights $\hat{\mathbf{W}}$ will be used to map embeddings for a specific classification result. For graph classification, the mapped embedding is usually a pooled embedding across all nodes, while for link prediction, a concatenated embedding of the two end nodes is often used.

**Principal Component Analysis (PCA)** is a classic feature reduction method to generate shorter but equally effective surrogate features based on original ones. Given a raw feature $\mathbf{X} \in \mathbb{R}^{N \times D}$, where $D$ is the feature dimension and $N$ is the data amount, it first applies an eigenvalue decomposition:

$$\frac{1}{D}\mathbf{X}^T\mathbf{X} = U\Sigma U^T \tag{2}$$

$U$ is a matrix containing all eigen vectors and $\Sigma$ is a diagonal matrix containing eigenvalues. To obtain a surrogate feature $\mathbf{X}_{D'} \in \mathbb{R}^{N \times D'}$ which has similar effectiveness to $\mathbf{X}$ but contains only $D'$ dimension ($D' \le D$), it takes the first $D'$ columns of $U$ and multi $\mathbf{X}$ with these reduced vectors:

$$\mathbf{X}_{D'} = \mathbf{X}U_{(:,D')} \tag{3}$$

The obtained matrix in the shape of $N \times D'$ is used as the desired $\mathbf{X}_{D'}$, which is further proved to have approximately the same power as $\mathbf{X}$ in Mohri et al. (2012). For the ease of expression, we write such a surrogate ability as:

$$\mathbf{X}_{D'} \overset{f}{=} \mathbf{X} \tag{4}$$

## 3 FOUNDATIONAL GNN ON PCA ALIGNED FEATURES

In this section, we will introduce our PCA feature alignment-based GFM in three steps. (1) First, we define the identical equivalence properties and explain how PCA alignment satisfies them(§3.1). (2) Based on these advantages, we secondly propose our Mini-GFM framework for PCA-aligned datasets (§3.2), which is simple yet proved to be empirically effective. (3) Thirdly, we make a theoretical analysis of the cross-domain generalization of our framework and prove its effectiveness in capturing latent space invariance across different domains' datasets (§3.3).

### 3.1 PCA SATISFIES IDENTICAL EQUIVALENCE FOR ZERO-SHOT GENERALIZATION

**Feature Generalization of Zero-Shot on Identical Graph Data** For ease of notation, we define $\mathcal{P}_n$ as the set of all permutation matrices in the shape of $n \times n$. Slightly different from Shen et al. (2024), we define the identical equivalence on GNN zero-shot generation into three specific properties. Formally, if a function $\hat{g}$ on graph features $\mathbf{X}$ satisfies:

(1) Node permutation equivalence: if an arbitrary permutation $\mathbf{P}_{\mathcal{V}} \in \mathcal{P}_N$ is applied to change the index order of the nodes, the set of node embeddings generated by the function $f$ is equivalent as changed before while only order permuted in the same way:

$$f(\mathbf{P}_{\mathcal{V}}\mathbf{X}) = \mathbf{P}_{\mathcal{V}}f(\mathbf{X}) \tag{5}$$

(2) Feature permutation invariance: if an arbitrary permutation $\mathbf{P_X} \in \mathcal{P}_D$ is applied to change the dimension order of the node feature, the set of node embedding generated by the function $f$ is invariant or expressively the same as before:

$$f(\mathbf{X}\mathbf{P_X}) = f(\mathbf{X}) \tag{6}$$

(3) Feature scaling invariance: if each feature dimension is arbitrarily scaled with a diagonal matrix $S$ with all diagonal values positive, the set of node embeddings generated by function $f$ is invariant:

$$f(\mathbf{S}\mathbf{X}) = f(\mathbf{X}) \tag{7}$$

While we notice that scaling invariance is easy to achieved by performing a normalization on $\mathbf{X}$ in prior, and adopting a graph neural network as $f$ also naturally satisfies node permutation equivalence, the feature permutation invariance is the hardest to achieve. If we could find an alignment function $g$ that satisfies both (1) node permutation equivalence and (2) feature permutation invariance, then a hierarchy GNN model $\hat{f} := f(g(\text{norm}(\mathbf{X})))$ could achieve all above identical invariance properties.

**Theorem 1. PCA reduced feature achieves Dimension Permutation Invariance:** *Given a feature* $\mathbf{X}$*, its arbitrary permuted features* $\mathbf{X}' = \mathbf{X}\mathbf{P_X}$ *with permutation matrix* $\mathbf{P_X} \in \mathbb{P}(D)$ *and a desired reduced dimension* $D'$*, after applying the PCA reduced respectively to* $\mathbf{X}_D$ *and* $\mathbf{X}'_D$*, the reduced features holds same approximate effectiveness as* $\mathbf{X}$*:* $f(\mathbf{X}'_{D'}) = f(\mathbf{X}_{D'})$*.*

*Proof.* (Sketch) When applying PCA alignment in Eq.3, by $\mathbf{X}\mathbf{P_X}\mathbf{P_X}^T U_{(:,D')} = \mathbf{X}U_{(:,D')}$, $\mathbf{P_X}$ is eliminated and resulted $\mathbf{X}_{D'}$ remains the same. For a fully formulated proof, please check in A.1. $\square$

**Theorem 2. PCA reduced feature satisfies Node Permutation Equivalence:** *Given a feature* $\mathbf{X}$*, its arbitrary permuted features* $\mathbf{X}' = \mathbf{P}_\mathcal{V}\mathbf{X}$ *with permutation matrix* $\mathbf{P}_\mathcal{V} \in \mathbb{P}(N)$ *and a desired reduced dimension, after applying the PCA reduced respectively to* $\mathbf{X}_D$ *and* $\mathbf{X}'_D$*, the reduced features satisfies the node permutation equivalent:* $\mathbf{P}_\mathcal{V}\mathbf{X}_{D'} = \mathbf{X}'_{D'}$*.*

*Proof.* (Sketch) When applying PCA alignment in Eq.2, by $\mathbf{X}^T\mathbf{P}_\mathcal{V}^T\mathbf{P}_\mathcal{V}\mathbf{X} = \mathbf{X}^T\mathbf{X}$ is eliminated and decomposed $U_{(:,D')}$ remains the same. Therefore $\mathbf{X}'_{D'} = \mathbf{P}_\mathcal{V}\mathbf{X}U_{(:,D')} = \mathbf{P}_\mathcal{V}\mathbf{X}_{D'}$ For a fully formulated proof, please check in A.2. $\square$

The above findings show that PCA is an ideal alignment function $g$ that we look for, and motivate us to build a graph foundation model framework with the hierarchy design mentioned above.

### 3.2 PCA-BASED GRAPH FOUNDATION MODEL FRAMEWORK

**PCA Feature Alignment**   We introduce our PCA feature alignment method to align all features $\mathbf{X}$ from a given dataset $\mathcal{D}$ with dimension $D$ to a fixed dimension $D'$. This alignment practically consists of three steps: **(a) Initialization**: If the given feature's dimension $D$ is less than $D'$, we concatenate it with itself repetitively until exceeds; if the given feature is none, we use all ones feature in shape of $N \times D'$ as initial feature directly. **(b) Normalization**: We take a normalization along the feature dimension. By doing so, we could mitigate the influence of the scaling $\mathbf{S}$ and satisfy the scaling invariance property. We note the normalized feature as $\hat{\mathbf{X}} := \text{norm}(\mathbf{X})$.**(c) PCA Computation**: If $\mathcal{D}$ is for node classification task, we use the all training nodes' features $\hat{\mathbf{X}}_{(\mathcal{V}_{\text{train}})}$ to calculate the $U$ matrix, while if for graph classification, we collect all training graphs' node features $\{\hat{\mathbf{X}}_G, \forall G \in \mathcal{D}_{(\text{train})}\}$ together to calculate $U$ for higher precision. Then, for every node $v \in \mathcal{D}$, we multiply its feature $\hat{\mathbf{X}}_{(v)}$ by the first $D'$ columns of $U$ and get the aligned feature $\mathbf{X}_{(v)_{D'}}$.

With the proposed PCA alignment method $g(\text{norm}(\mathbf{X})) = \mathbf{X}_{D'}$, we introduce our Mini-GFM framework at a higher level across pretrain datasets $\mathcal{D}^{\text{pre}}$ and generalization datasets $\mathcal{D}^{\text{gen}}$:

**Mini-GFM Pipeline**   Our graph foundation framework generally consists of three steps, which are illustrated in Figure 2. In the **Step I**, we apply the above PCA alignment for all graphs (including the test graphs) across all pretrain datasets $\mathcal{D}^{\text{pre}}$. In the **Step II,** we train both the core graph model, a relatively large graph neural network to learn graph embedding for all pretrain datasets $\mathcal{D}^{\text{pre}}$, and the downstream task layers, which are independent for each dataset $\mathcal{D}_i \in \mathcal{D}^{\text{pre}}$. In **Step III**, when generalizing to a new dataset $\mathcal{D}_j \in \mathcal{D}^{\text{gen}}$, we first align the graph data and then feed it forward to

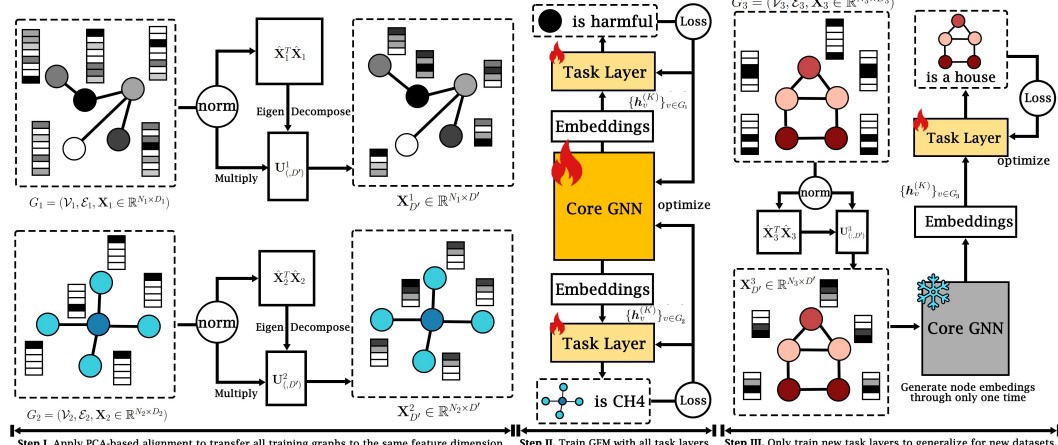

Figure 2: Our Mini-GFM framework could be summarized into a three-step pipeline. In Step I, we align features for every training dataset. In Step II, we use aligned graph datasets to train both our core GNN model and downstream task layers. In Step III, when generalizing to new datasets, we forward-align the graph to obtain a zero-shot embedding and only train a downstream task layer.

| Terms | Train Alignment | | Generalize Alignment | |
|---|---|---|---|---|
| Complexity | $O(D_1^2(N_1 + D_1) + N_1 D_1 D')$ | | $O(D_2^2(N_2 + D_2) + N_2 D_2 D')$ | |
| Terms | Core Train | Core Generalize | Down. Train | Down Generalize |
| Complexity | $O(T_{pre}D'|\mathcal{E}_1|L)$ | $O(D'|\mathcal{E}_2|L)$ | $O(T_{pre}D'C_1)$ | $O(T_{gen}D'C_2)$ |

Table 1: **Mini-GFM complexity.** PCA alignment is a one-time cost per dataset that is *quadratic in feature dimension* and *linear in sample size* and projection size. After alignment, transfer is *edge-linear* for the core GNN pass and *class-linear* for fitting the shallow head. *Takeaway:* zero-shot adaptation is dominated by a single edge-linear forward plus a small linear head; no per-edge surrogates or LLM preprocessing.

the core GNN to directly obtain a statistical embedding. Depending on the specific task, we only train a downstream linear layer $\hat{\mathbf{W}}^j$ to map the embedding for classification, while the core model is free of learning and serves as a zero-shot embedding generator. This pipeline aligns with the idea of designing a foundation model easy for generalization.

**Complexity Analysis** The complexity of our method could be analyzed in five terms: alignment complexity, core GNN model training complexity & generalization complexity, downstream training complexity & generalization complexity. Without loss of generality, we assume: take $T_{pre}$ epochs for pretraining on a node classification dataset $G_1 = (\mathcal{V}_1, \mathcal{E}_1, \mathbf{X}_1)$ with $N_1$ nodes, $D_1$ feature dimensions and $C_1$ classes; $T_{gen}$ for generalizing to a node dataset $G_2 = (\mathcal{V}_2, \mathcal{E}_2, \mathbf{X}_2)$ with $N_2$ nodes, $D_2$ feature dimensions and $C_2$ classes; $D'$ is also the hidden size of the $L$ layer core model. The time complexity expression is listed in Table 1 To special note, when we zero-shot generalize for $G_2$, we only need to afford a $O(D'|\mathcal{E}_2|L + T_{gen}D'C_2)$ complexity to fully generalize our model to an unseen graph data. We also compare with other baselines on complexity to generalize zero-shot graph presentations in Table 2, which indicates our methods are efficient in practice.

| Method | Complexity |
|---|---|
| STAGE Shen et al. (2024) | $O(|\mathcal{E}_2|D_2^2 D' + |\mathcal{E}_2|D' + N_2 D'^2)$ |
| GraphAny Zhao et al. (2024) | $O(D' D_2 |\mathcal{E}_2| + D' D_2^2 N)$ |
| Mini-GFM (Ours) | $O(D'|\mathcal{E}_2|L + D_2^2(N_2 + D_2) + N_2 D_2 D')$ |

Table 2: **Transfer-time comparison.** *STAGE* is dominated by *per-edge surrogate construction* (cost grows with edges and the square of feature dimension). *GraphAny* performs a GNN pass and solves least squares problem once for every linearGNN. *Mini-GFM* performs one PCA and a single *edge-linear* GNN pass, then fits a shallow head. *Takeaway:* on sparse graphs or moderate feature dimensions, Mini-GFM and GraphAny scales better and adapts with minimal extra computation.

To better illustrate the motivation of our framework design, we additionally present key points against some crucial questions as follows:

**Why only zero-shot generalize the core model but not the downstream task layer?** In this work, we do not focus on these due to the fact that (1) mostly, downstream labels are not strictly related to the input graph data itself and are uncontrollable. (2) Downstream task learning is in linear complexity, i.e., $O(DC)$, and has no relation to the scale of the graph, which is relatively smaller compared with typical graph learning, i.e., $O(LD^2|\mathcal{E}|)$, especially in the face of large-scale data.

**Does the pre-processing of up-sampling original features by repeating break the expressiveness?** It's easy to prove that identical invariance under such up-sampling could still be achieved by PCA alignment. The sketch of the proof is that the upsampling of the features could be viewed as another matrix $\mathbf{P} = [\mathbf{I} \dots \mathbf{I}]^T \in \mathbb{R}^{(r*d) \times d}$. Similar to the permutation matrix, since $\frac{1}{r*d}\mathbf{P}^T\mathbf{P} = \frac{1}{r}\mathbf{I}^T\mathbf{I}$, this term will be eliminated again during the PCA alignment.

**Why not directly sample a series of permutation matrices and use them for data augmentation?** In the expectation, applying sampled permutation matrices is equivalent to training over $\Sigma_{\mathbf{P} \in \mathbb{P}}\mathbf{XP}$. Therefore, for a node $v$ its every feature is expected to equal to $\sum_{i \in [D]} \mathbf{X}_{(v,i)}$, which is indeed smoothed into the sum of all $v$'s features. In other words, the feature matrix would be reduced to a one-dimensional feature containing every node's feature sum, leading to huge information loss.

## 3.3 PCA Satisfies Latent Equivalence for Zero-shot Generalization

As demonstrated in previous sections, holding an equivariant representation of identical graph data is essential for designing generalizable graph foundation models. However, this fundamental property still can't guarantee effective generalization on non-identical datasets across domains. Observing from CV and NLP pretrained models, we found that it is most important to find invariance for generalization across domains like spatial relations or sequential relations. However, it's not straightforward to find the invariance that holds among graph features. Therefore, we make an assumption that there's a latent space that may be shared across datasets from two domains:

**Assumption 3.** Let $i \in \{1, 2\}$ index two graph domains containing graph sample $G_i = (\mathbf{A}_i, \mathbf{X}_i)$. For each domain $i$, we assume the node features could be decomposed as:

$$\mathbf{X}_i = \mathbf{H}B_i + E_i \tag{8}$$

where $\mathbf{H} \in \mathbb{R}^{N_i \times r}$ is a shared rank-$r$ latent matrix, $B_i \in \mathbb{R}^{r \times D}$ is a domain-specific linear map, and $E_i$ is feature noise.

We also assume that there exist some matrix $\mathbf{M}_i \in \mathbb{R}^{r \times D'}$ that the label $Y_i$ follows:

$$Y_i = \sigma(\mathbf{A}_i\mathbf{H}\mathbf{M}_i) + \Xi_i \tag{9}$$

where $\sigma : \mathbb{R} \to \mathbb{R}$ is an element-wise function and satisfies L-Lipschitz, and $\Xi_i$ is label noise.

Therefore, the expected covariance matrix of $\mathbf{X}_i$ satisfies:

$$\Sigma_i := \mathbb{E}[\frac{1}{n_i}\mathbf{X}_i^T\mathbf{X}_i] = B_i^T\Sigma_\mathbf{H}B_i + \Sigma_{E_i}, \quad \Sigma_\mathbf{H} := \frac{1}{n_i}\mathbf{H}^T\mathbf{H}$$

Without loss of generalization, we note the $\lambda_1(\Sigma_i) \geq \cdots \geq \lambda_f(\Sigma_i)$ as eigenvalues, and define the eigengap as $\delta_i$:

$$\delta_i := \lambda_r(\Sigma_i) - \lambda_{r+1}(\Sigma_i) > 0 \tag{10}$$

We note the covariance matrix on data sample as $\hat{\Sigma}_i := \frac{1}{n_i}\mathbf{X}_i^T\mathbf{X}_i$, and let $U_i \in \mathbb{R}^{D \times D'}$ contain the top $D'$ eignvectors ($D' \geq r$). Define the perturbation $\Delta_i := \hat{\Sigma}_i - \Sigma_i$, with spectral norm $\epsilon_i := ||\Delta_i||_2$

Under a shared GCN layer with linear weight $\mathbf{W} \in \mathbb{R}^{D' \times D'}$, we write the aggregated embedding as:

$$Z_i := \mathbf{A}_i\mathbf{X}_iU_i\mathbf{W}$$

**Theorem 4. PCA alignment ensures generalizable embeddings** *For each domain $i$, there exist a matrix $\mathbf{M}_i$ such that*

$$||Z_i - \mathbf{A}_i\mathbf{H}\mathbf{M}_i||_F \leq ||\mathbf{W}||_2\kappa_i(C_1\frac{\epsilon_i}{\delta_i}||\mathbf{H}||_F + C_2||E_i||_F) \tag{11}$$

*where $\kappa_i$ is the conditioning of the coordinate transform between the PCA subspace and the latent span, and $C_1, C_2 > 0$ are absolute constants.*

*Consequently, for any linear head $\hat{\mathbf{W}}_i \in \mathbb{R}^{D' \times c}$,*

$$||\sigma(Z_i)\hat{\mathbf{W}}^i - Y_i||_F \leq L||\hat{\mathbf{W}}^i||_2||Z_i - \mathbf{A}_i\mathbf{H}\mathbf{M}_i||_F + ||\sigma(\mathbf{A}_i\mathbf{H}\mathbf{M}_i)\hat{\mathbf{W}}^i - Y_i||_F$$

$$\leq L||\hat{\mathbf{W}}^i||_2||\mathbf{W}||_2(C_1\frac{\epsilon_i}{\delta_i}||\mathbf{H}||_F + C_2||E_i||_F) + ||\sigma(\mathbf{A}_i\mathbf{H}\mathbf{M}_i)\hat{\mathbf{W}}^i - Y_i||_F \tag{12}$$

*Proof.* By Assumption.3, the signal eigenspace of $\Sigma_i$ is $r$-dimensional with eigengap $\delta_i$. Davis-Kahan's sin$\Theta$ theorem Stewart & guang Sun (1990); Yu et al. (2015) yields:

$$||\sin\Theta(\text{col}(U_i), \text{col}(B_i^T))||_2 \leq \frac{||\Delta_i||_2}{\delta_i} = \frac{\epsilon_i}{\delta_i}$$

Since $\text{col}(U_i)$ is within angle $\eta := \frac{\epsilon_i}{\delta_i}$ of $\text{col}(B_i^T)$, there exists a matrix $R_i \in \mathbb{R}^{p \times r}$ with bounded pseudo-inverse norm $||R_i^+||_2 = \kappa_i$ such that:

$$||\mathbf{X}_iU_iR_i - \mathbf{H}||_F \leq C_1\eta_i||\mathbf{H}||_F + C_2||E_i||_F \tag{13}$$

for absolute constants $C_1, C_2$ Stewart (1998).

Then, by setting a basis matrix $\mathbf{M}_i := R_i^+\mathbf{W}$, the distance between the aggregated embedding and the latent feature space could be expressed as:

$$Z_i - \mathbf{A}_i\mathbf{H}\mathbf{M}_i = \mathbf{A}_i(\mathbf{X}_iU_i\mathbf{W} - \mathbf{H}R_i^+\mathbf{W})$$

$$= \mathbf{A}_i((\mathbf{X}_iU_iR_i - \mathbf{H})R_i^+\mathbf{W})$$

Therefore,

$$||Z_i - \mathbf{A}_i\mathbf{H}\mathbf{M}_i||_F \leq ||\mathbf{A}_i||_2||\mathbf{W}||_2\kappa_i||\mathbf{X}_iU_iR_i - \mathbf{H}||_F$$

Notice the adjacency matrix is normalized, which is 1-Lipschitz, and recall Eq. 13

$$||Z_i - \mathbf{A}_i\mathbf{H}\mathbf{M}_i||_F \leq ||\mathbf{W}||_2\kappa_i||\mathbf{X}_iU_iR_i - \mathbf{H}||_F$$

$$\leq ||\mathbf{W}||_2\kappa_i(C_1\frac{\epsilon_i}{\delta_i}||\mathbf{H}||_F + C_2||E_i||_F) \quad \text{(recall Eq.(13))} \tag{14}$$

Since $\sigma$ is assumed to be $L$-Lipschitz, we further have

$$||\sigma(Z_i)\hat{\mathbf{W}}^i - Y_i|| \leq ||\sigma(Z_i)\hat{\mathbf{W}}^i - \sigma(\mathbf{A}_i\mathbf{H}\mathbf{M}_i)\hat{\mathbf{W}}^i||_F + ||\sigma(\mathbf{A}_i\mathbf{H}\mathbf{M}_i)\hat{\mathbf{W}}^i - Y_i||_F$$

$$\leq L|\hat{\mathbf{W}}^i||_2||Z_i - \mathbf{A}_i\mathbf{H}\mathbf{M}_i||_F + ||\sigma(\mathbf{A}_i\mathbf{H}\mathbf{M}_i)\hat{\mathbf{W}}^i - Y_i||_F \tag{15}$$

$\square$

This theorem offers the insight that the PCA alignment helps to map the raw features toward the shared latent features (as Eq. 13). Meanwhile, an independent downstream linear layer not only helps to map the representation toward the specific task labels, but also offers flexibility in reducing the misaligned basis introduced in shared weights. In other words, with the PCA alignment and trainable downstream layer, our Mini-GFM framework is able to capture latent invariance across datasets from different domains with zero-shot graph representation generalization and only linear complexity for task adaptations.

## 4 EXPERIMENTS

**Datasets** We use 8 node classification datasets and 8 graph classification datasets in different domains from PyG (PyTorch Geometric) Fey & Lenssen (2019). Details of them are shown in Table 4. Additionally, we also test our trained GFM on two big datasets with zero-shot generalization to further validate our method's effectiveness.

| Node Data. | Orig. | Mess. | Repeat | GraphAny | Ours | Graph Data. | Orig. | Mess. | Repeat | Ours |
|---|---|---|---|---|---|---|---|---|---|---|
| Amazon-C | 0.914 | 0.889 | 0.879 | 0.888 | **0.891** | DD | 0.691 | 0.555 | 0.675 | **0.682** |
| Amazon-P | 0.931 | 0.932 | 0.926 | 0.928 | **0.936** | AIDS | 0.988 | 0.199 | 0.978 | **0.983** |
| Cora-ML | 0.813 | 0.737 | **0.795** | 0.615 | 0.791 | ENZYMES | 0.460 | 0.167 | 0.374 | **0.489** |
| PubMed | 0.874 | 0.838 | 0.842 | **0.878** | 0.844 | MUTAG | 0.790 | 0.555 | 0.765 | **0.773** |
| CiteSeer | 0.743 | **0.723** | 0.669 | 0.592 | 0.684 | PROTEINS | 0.754 | 0.598 | 0.673 | **0.733** |
| Airports | 0.565 | 0.517 | 0.534 | 0.405 | **0.548** | IMDB-B | 0.661 | 0.500 | 0.664 | **0.664** |
| KarateClub | 0.800 | 0.875 | 0.875 | 0.875 | **0.900** | COLLAB | 0.698 | 0.155 | 0.656 | **0.679** |
| Wiki | 0.742 | 0.673 | **0.724** | 0.461 | **0.724** | REDDIT-B | 0.803 | 0.500 | 0.684 | **0.759** |

Table 3: Performance comparison between the original fully trained GNN with our GFM framework.

**Experiment Setting**  To evaluate the empirical effectiveness of our method, we divide the 16 datasets into 4 groups. Every time we take 3 groups, i.e., 6 node classification datasets and 6 graph classification datasets, as pretraining datasets to pretrain our core GFM model with independent downstream task layers for 1000 epochs. Afterwards, for the remaining test group with 2 node datasets and 2 graph datasets, we directly forward them to the pretrained core model only once, and store the generated embeddings. Then, for each of them, we only train the corresponding downstream linear layer for 2000 epochs on the training set and evaluate the test set prediction accuracy as our final results. For each dataset, we use 50% of the nodes/graphs as training data, 20% as validation data, and 30% as test data. Every 2 lines in Table3 represent result 1 generalization group's test results. For the GNN core model, we applied a 4-layer GCN model with a hidden size of 512. The align dimension $D'$ is set to 1024.[2]

**Baseline**  To demonstrate the effective performance of Mini-GFM we compare with two kind of baselines: (1) identical equivalence generalization baseline: GraphAnyZhao et al. (2024)[3] and (2) ablation baselines: "Repeat" that align features by simply copying raw features along the dimension and "Message" that apply non-parameter aggregation on GCN layer with only message passing. Due to the task limitation of GraphAny and the high complexity of STAGE, we perform GraphAny only for node classification datasets and STAGE only for graph classification datasets, respectively. Ablation baselines are performed in a similar pipeline, with only alignment replaced for "Repeat" and the core model only aggregate raw features for "Message".

**Main Performance Analysis**  As Table 3 shows, our zero-shot generalization results are competitive compared with originally fully trained GNN models, and mostly outperform both identical equivalence baselines and ablation baselines. This not only demonstrates the effectiveness of our foundation model framework, but also reveals the sufficiency of PCA as feature alignment method. In 4 datasets, our method's performance is even better than the original GNN's performance, which suggests that training foundational large graph models may bring potential benefits of increasing the capacity of learnt graph representation.

Among baselines, we notice that GraphAny slightly outperforms ours in Pumbed, and in Amazon-C&P it also achieves similar performance. This is impressive especially considering their end-to-end prediction design. However, when the feature dimension grows higher in datasets like Cora-ML, CiteSeer and Airports, its insufficiency appears apparently. This may potentially due to its basis classifications are built on rough approximation models. Though these statistical models offer a lower complexity, their inaccuracy may lead the final aggregated prediction to bad performance. Besides, it's also surprising to find that the simple "Message" could also achieve considerable performance in a few high dimension datasets. After analysis, we suggest that it may due to the high uniqueness of features in these datasets, especially in CiteSeer where $D$ is even higher than the node amount. This implies that some nodes may occupy unique features and are easy to be classified even without complex weight updates. While in most cases, its performance is still relatively low.

**Large Dataset Test**  To further validate our method's capacity for large scale application, we test the performance on a large dataset Amazon-2M from Open Graph Benchmark, which is a product

---

[2]Code for reproduction could be found in the following anonymous link `https://anonymous.4open.science/r/MiniGFM/`.

[3]For STAGEShen et al. (2024), we couldn't find available public implementation, and our replication finds that the complexity relates to $O(|\mathcal{E}|D^2 D')$ is extremely too high for our experiments setting.

| Node Data. | $|\mathcal{V}|$ | $|\mathcal{E}|$ | $D$ | $C$ | Graph Data. | $|\mathcal{G}|$ | $|\mathcal{V}|_{avg}$ | $|\mathcal{E}|_{avg}$ | $D$ | $C$ |
|---|---|---|---|---|---|---|---|---|---|---|
| Cora-ML | 2,708 | 10,556 | 2879 | 7 | DD | 1,178 | 284.3 | 715.7 | 89 | 2 |
| PubMed | 19,717 | 88,648 | 500 | 3 | AIDS | 2,000 | 15.7 | 16.2 | 42 | 2 |
| Amazon-C | 13,752 | 491,722 | 767 | 10 | ENZYMES | 600 | 32.6 | 62.1 | 21 | 6 |
| Amazon-P | 7,650 | 238,162 | 745 | 8 | PROTEINS | 1,113 | 39.1 | 72.8 | 4 | 2 |
| CiteSeer | 3,327 | 9,104 | 3,703 | 6 | MUTAG | 4,337 | 30.3 | 30.8 | 14 | 2 |
| Airports | 1,190 | 13,599 | 1,190 | 3 | IMDB-B | 1,000 | 19.8 | 96.5 | 0 | 2 |
| KarateClub | 34 | 156 | 34 | 4 | COLLAB | 5,000 | 74.5 | 2457.8 | 0 | 3 |
| Wiki | 28,281 | 185,504 | 128 | 2 | REDDIT-B | 2,000 | 429.6 | 497.8 | 0 | 2 |

Table 4: Details on experiments datasets.

| Datasets | $|\mathcal{V}|$ | $|\mathcal{E}|$ | $D$ | $C$ | Orig. | Ours | $T_{Orig.}$ | $T_{ours}$ |
|---|---|---|---|---|---|---|---|---|
| Amazon-2M | 2,449,029 | 61,859,140 | 100 | 47 | 0.603 | 0.601 | 306min | 204min |

Table 5: Description and Performance on Large Graph Dataset

purchase network containing 2 million nodes. In this task we first train our models on the 16 small datasets, and then test its generalization on the Amazon-2M datasets. Adopting the division strategy applied in ClusterGCNChiang et al. (2019), for both the original GCN and our Mini-GFM, we first randomly divide the original graph into 20 subgraphs and then train downstream layer/ GCN respectively. We set the train epoch as 2000 and train-valid-test ratio as 0.3:0.2:0.5. The performance and recorded time are shown in the Table 5. By using a relatively shorter generalization time, our GFM framework could adapt for a similar performance as a fully trained GCN.

## 5 RELATED WORKS

**Graph Foundation Models**   Most existing GFM works could be generally categorized as two groups:(1) Domain-specific GNNShoghi et al. (2024); Sypetkowski et al. (2024); Wang et al. (2024); Klaser et al. (2024); Jia et al. (2025); Li et al. (2025) develops foundation models for a specific domain, e.g., the molecular graph, which naturally shares a same feature space; (2) Text-attributed GNNLiu et al. (2024); Xia et al. (2024); Hou et al. (2024); Huang et al. (2023); Kong et al. (2025); Tang et al. (2024); He et al. (2025) designs GNN with language model encoder to map context description to node features. Besides these two main trends, there are also works on other directions like surrogate featuresZhao et al. (2024); Shen et al. (2024), structure alignmentYu et al. (2025b), and downstream task generalization Bevilacqua et al. (2025); Sun et al. (2023).

To be specifically noted, there are also preliminary works Yu et al. (2024; 2025a) that have applied feature dimension reduction, such as PCA or SVD reduction, to align features before building cross-domain graph models. However instead of analyzing the effectiveness of PCA itself, these works just assume it as a pre-processing tool for fixing the same feature length across datasets while primarily rely on the effectiveness of graph tokens and graph transformers. On the contrary, our findings emphasize valuable properties introduced by PCA itself.

## 6 CONCLUSION

We study transfer for graph foundation models (GFMs) under heterogeneous node attributes and topology across datasets. Our main finding is that PCA is a sufficient and efficient feature alignment for GFMs. We prove that PCA-aligned features satisfy two invariance properties central to zero-shot transfer on identical datasets, and establish a representation bound when the source and target share a latent feature space. Building on these, we propose **Mini-GFM**, which pretrains a single core GNN across multiple datasets and, at transfer time, requires only PCA on the target dataset plus a shallow task-specific linear head. Empirically, Mini-GFM delivers competitive zero-shot performance while avoiding heavy auxiliary pipelines such as LLM-based surrogates or surrogate feature graphs through extensive experiments. Its alignment and transfer steps are simple and low-cost, making it practical at scale. Future directions include (i) designing alignment beyond PCA (e.g., whitening, CCA-like variants) with formal guarantees, (ii) combining structure-aware alignment that couples feature and topology statistics, (iii) empirically extensions to regression, link prediction, and dynamic graphs, and (iv) tighter bounds that account for higher-order graph operators and stronger distribution shifts.

## 7 ETHICS STATEMENT

We use only public benchmark datasets (e.g., PyG/OGB) under their licenses; no new human data were collected, and we report aggregate results without attempting to identify individuals. To our knowledge, the datasets contain no personally identifiable information, and we will not redistribute data whose licenses prohibit it. Because our approach enables rapid zero-shot transfer, it could be applied to sensitive graphs; any such use should comply with law, institutional review, and data-governance policies, and it should include checks for group-wise performance, calibration, and distribution-shift sensitivity before deployment in high-stakes contexts. Our pipeline avoids LLM-based preprocessing, which reduces compute and limits propagation of text-derived biases.

## 8 REPRODUCIBILITY STATEMENT

We provide complete details to facilitate replication. The formal description of our PCA alignment, Mini-GGM framework and training procedures are posed in Section 3 and Section 4, including datasets, class partitions, evaluation process. 8 node datasets and 8 graph datasets choices are also described in the Section 4. Baselines and comparison settings are enumerated alongside our methods to ensure parity. Additional implementation details are provided in the appendix. Our anonymous code repository is linked in the main paper for reproduction.

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

## A PROOF

### A.1 PROOF FOR THEOREM 1

*Proof.* For $\mathbf{X}' = \mathbf{P_X X}$, we note its eigenvalue decomposition is expressed as:

$$\mathbf{P_X}^T \mathbf{X}^T \mathbf{X} \mathbf{P_X} = U'\Sigma'U'^T \tag{16}$$

recall **??**, we have the two decomposition equal as:

$$\mathbf{P_X}^T U \Sigma U^T \mathbf{P_X} = U'\Sigma'U'^T \tag{17}$$

since $\mathbf{P_X}$ is still an orthogonal matrix, it could be viewed as another decomposition of $\mathbf{X}'^T \mathbf{X}'$. According to the properties of eigenvalue decomposition, the eigenvalue matrix $\Sigma'$ is the same as $\Sigma$, and therefore $\mathbf{P_X}^T U$ could be viewed functionally equal to $U'$ (both in the space of possible decomposition vectors). In other words, for a desired reduced dimension $D'$:

$$\begin{aligned} \mathbf{X}'_{D'} &\overset{f}{=} \mathbf{X}'\mathbf{P_X}^T U_{(:,D')} \\ &= \mathbf{X}\mathbf{P_X}\mathbf{P_X}^T U_{(:,D')} \\ &= \mathbf{X}U_{(:,D')} = \mathbf{X}_{D'} \end{aligned} \tag{18}$$

Therefore $f(\mathbf{X}_{D'}) = f(\mathbf{X}'_{D'})$ regardless of whatever permutation $\mathbf{P_X}$ is taken, and they always hold approximate same expressiveness as $f(\mathbf{X})$. $\square$

### A.2 PROOF FOR THEOREM 2

*Proof.* For $\mathbf{X}' = \mathbf{P_X X}$, we note its eigenvalue decomposition is expressed as:

$$\mathbf{X}^T \mathbf{P}_{\mathcal{V}}^T \mathbf{P}_{\mathcal{V}} \mathbf{X} = U'\Sigma'U'^T \tag{19}$$

Since permutation matrix $\mathbf{P}_{\mathcal{V}}$ is also a orthogonal matrix, we have:

$$\mathbf{X}^T \mathbf{X} = U \Sigma U^T = U'\Sigma'U'^T \tag{20}$$

Note in implementation, when the matrix is exactly the same, the eigenvectors are always calculated the same as well. Therefore, we have:

$$\begin{aligned} U &= U' \\ \mathbf{X}'_{D'} &= \mathbf{X}'U'_{(:,D')} \\ &= \mathbf{P}_{\mathcal{V}}\mathbf{X}U_{(:,D')} \\ &= \mathbf{P}_{\mathcal{V}}\mathbf{X}_{D'} \end{aligned} \tag{21}$$

$\square$

## B    IMPLEMENTATION DETAILS

Here we provide further details of our implementation. For all experiments, we use Adam as our optimizer and set the training rate as 0.001. We adopt CrossEntropy as our loss criterion. To perform the experiments we use a RTX 6000 Ada on a linux server. All activation layers in models we use are ReLU layer. Details of baselines' setting are also stated below:

**GraphAny**  Follow the official implementation provided in `https://github.com/DeepGraphLearning/GraphAny`, we set the channel to 5, node-label examples as 5, hidden size as 128, entropy as 1, attention temperature as 5 and mlp layer as 2.

**Message GCN**  We apply the SimpleConv implementation in torch geometric as the message-passing layer. We use 4 layers and finally concatenate all hidden together as final embedding.

**Original GCN**  We use 3 GCN layers with hidden size as 256 and finally concatenate all hidden together as final embedding.

