# OpenReview forum: "PCA Feature Alignment is Sufficient for Building Graph Foundation Models"
_ICLR.cc/2026/Conference — Submitted to ICLR 2026_

### Official Review · Reviewer_dqxB · 2025-10-30

**Soundness:** 2
**Presentation:** 2
**Contribution:** 2
**Rating:** 2
**Confidence:** 4

**Summary:**

This paper focuses on the challenge of zero-shot generalization of Graph Foundation Models (GFMs) across datasets. Addressing issues such as heterogeneous node features, distinct topological structures of graph data from different domains, and the limitations of traditional methods, it proposes to use Principal Component Analysis (PCA) as a feature alignment tool and constructs the Mini-GFM framework. Through experiments on node classification and graph classification tasks across multiple domains, the paper verifies the framework's competitiveness in zero-shot scenarios while reducing preprocessing costs, providing a new approach for efficient generalization of GFMs.

**Strengths:**

1. Graph Foundation Models represent an important cutting-edge direction in the field of graph learning.

2. The experiments use 16 datasets for node classification and graph classification across different domains, including both small-scale standard datasets and the large-scale Amazon-2M dataset, showing a relatively rich selection of datasets.

**Weaknesses:**

- The paper proposes that when the original feature dimension D is smaller than the target alignment dimension D’, the feature is repeatedly concatenated until its dimension exceeds D’ for processing. **However, it fails to consider the information redundancy that may be introduced by repeated concatenation of high-dimensional features.**

- Although the paper demonstrates the advantage in training time on the Amazon-2M dataset, it does not analyze the computational complexity of PCA alignment in ultra-large-scale high-dimensional feature graphs (e.g., feature dimension D>10000, number of nodes exceeding 10 million). The core steps of PCA involve the calculation of the feature covariance matrix; when D is extremely large, the storage of the covariance matrix and eigenvalue decomposition will face memory and time bottlenecks. **Such high-dimensional graph data are common in practical industrial scenarios, and the paper does not explain how to solve this efficiency problem.**

- The paper assumes that datasets from different domains share a low-rank latent matrix **H** (Eq. 8), but does not verify the rationality of this assumption through experiments.

- Theorem 1 aims to prove the invariance of PCA to feature permutation. In the proof process, it mentions “**X’ =P_X·X**”, but according to the definition in the previous text, the feature permutation transformation should be “**X’ =X·P_X**” (Eq. 6), and there is **an error in the order of matrix multiplication here**. Additionally, there is a **citation error** in the proof (Line 618 of the paper).

- The paper does not analyze the generalization performance degradation pattern of Mini-GFM when the latent spaces of datasets differ significantly (e.g., molecular graphs and social network graphs), which makes me concerned about its practicality.

- The paper only sets two ablation baselines: "Repeat" and "Message", and does not conduct a sensitivity analysis on the key parameters of PCA alignment (e.g., the selection of target dimension).

- The paper only verifies node classification and graph classification tasks, and **does not involve common tasks in the field of graph learning such as link prediction and graph generation**.

- The experimental results are not marked with **standard deviations or confidence intervals**, making it impossible to judge the stability of the experimental results.

**Questions:**

See weaknesses.

---

### Official Review · Reviewer_kSxk · 2025-10-31

**Soundness:** 2
**Presentation:** 2
**Contribution:** 2
**Rating:** 2
**Confidence:** 4

**Summary:**

The paper argues for the use of PCA to align the feature space of different graphs, so that a single backbone model can be trained that will inductively generate embeddings for a new graph. These inductively generated embeddings then can be used directly for downstream tasks, such as node/graph classification/regression, and link prediction. This is overall, a novel idea.
The paper also presents a supporting theoretical argument for PCA being a good candidate for feature alignment.
Some evaluation has been done on this idea in this paper, but I find the evaluation severely lacking.

**Strengths:**

- The idea of using simple PCA for inductive feature alignment across datasets is novel.
- The paper presents theoretical arguments to back the use of PCA.

**Weaknesses:**

- The paper classifies its presented method as "zero-shot" when it clearly needs labels to learn a classification/regression head. The term "inductive" is more standard to be used in this scenario. "Zero-shot" indicates a lack of need for labels to generalize to a new task.
- Handling scale invariance: The proposed method only handles the scale invariance of the type $SX$, but fails to mention the case $XS$. If this case is not realistic, a textual argument should be made as to why not.
- Very few evaluations were done, and they do not include any standard deviation measure. For these problems, I would expect a paper to have a results table something like Table 5 in GraphAny.
- The paper could additionally be improved with an evaluation on the task of link prediction (though my other concerns are far more important than this one).
- The way of using bold text within table 3 is very different from what is standard. It does not recognize when a simple GNN beats the proposed method. Additionally, the fact that bold is not used for GNN is not mentioned anywhere. This will confuse readers into thinking the proposed methods are always better than a GNN.
- Overall, the paper spends too much space explaining classic concepts of GCNs and PCA, and too little space on doing actual evaluation.

**Typos**:
- Line 053 - "presentations" -> "representations"
- Line 135 - "multi" should be "multiply"
- Linee 158 - "Node permutation equivalence" -> "Node permutation equivariance" (equivariance is the standard term to use here)

**Questions:**

- Normalization: Why form of normalization is used? It is not mentioned near line 201.

---

### Official Review · Reviewer_gzW1 · 2025-10-31

**Soundness:** 1
**Presentation:** 2
**Contribution:** 1
**Rating:** 2
**Confidence:** 5

**Summary:**

The authors focus on addressing the issue of inconsistent feature dimensions across different graph datasets in Graph Foundation Models (GFMs). They propose that Principal Component Analysis (PCA) is sufficient for aligning graph features. Based on this, they introduce a simple GFM method to achieve cross-dataset generality.

**Strengths:**

- Graph foundation models is a valuable research area.
- Investigating how to align features across different graph datasets is worthwhile.
- Restating the properties of the PCA holds some value.

**Weaknesses:**

- The theoretical content appears somewhat dated or previously established. In my view, the two properties of PCA emphasized in the paper ("Dimension Permutation Invariance" and "Node Permutation Equivalence") are already well-recognized in the academic community. As noted in S3, the paper's contribution is primarily a restatement of PCA's properties rather than a new proof.
- In my opinion, the theory presented does not adequately support the claim that "PCA Feature Alignment is Sufficient for Building Graph Foundation Models." The proposed theory relies on Assumption 3, which posits that different graph datasets share latent features. However, this assumption may not hold for graph datasets. For instance, some graph datasets have no feature, which clearly does not satisfy Equation 8.
- The experiments conducted in the paper seem insufficient and could benefit from greater rigor. For a graph foundation model, it would be appropriate to test the proposed method in few-shot scenarios. The current experimental setup is not typical for GFMs (e.g., the division of pre-training and testing graph data, as well as the data split ratios).
- The authors claim that PCA can align graph features. To further validate this view, I suggest supplementing experiments that transfer from citation datasets to molecular datasets (such as BBBP, Tox21, QM9, ZINC) to further substantiate this view. This would be particularly valuable in strengthening the argument.
- The proposed method lacks novelty. Existing works already employ PCA to unify features across different graph data, such as RiemannGFM [1], AnyGraph [2], All-In-One [3], and others.
- Existing work, such as FUG [4], has already analyzed the relationship between PCA and graph feature unification. This paper could benefit from discussing them.

[1] RiemannGFM: Learning a Graph Foundation Model from Riemannian Geometry, WWW-25.

[2] AnyGraph: Graph Foundation Model in the Wild, Arxiv-24

[3] All in One: Multi-task Prompting for Graph Neural Networks, KDD-23

[4] FUG: Feature-Universal Graph Contrastive Pre-training for Graphs with Diverse Node Features, NeurIPS-24

**Questions:**

Please see the weaknesses.

---

### Official Review · Reviewer_sfdt · 2025-11-01

**Soundness:** 2
**Presentation:** 2
**Contribution:** 1
**Rating:** 2
**Confidence:** 4

**Summary:**

This paper directly uses principal component analysis to align feature from different graphs and proposes the Mini-GFM framework. The method is validated across multiple datasets, demonstrating its effectiveness. Compared to existing methods, the proposed method is efficient and scalable for large graph data.

**Strengths:**

S1. The solved problem is valuable.

﻿
S2. The proposed method is validated on diverse datasets, including both node and graph classification tasks.

﻿
S3. The authors provide a detailed time complexity analysis.

**Weaknesses:**

W1. The proposed method is too simple and lacks novelty. Many existing works (e.g., MDGFM) have already used PCA to align features across different datasets.

W2: The experimental evaluation lacks comparison with several state-of-the-art graph foundation models, such as MDGPT and MDGFM.

W3. The paper lacks few-shot learning experiments. Few-shot generalization is an important capability for graph foundation models.

**Questions:**

Q1. Why does the proposed method outperform the original fully trained GNN on some datasets (e.g., KarateClub)?

Q2. How does the proposed method perform when pre-trained on only a single dataset? Can the proposed method truly acquire extensive graph knowledge through multi-domain pre-training?

---

### Author Response · Authors · 2025-12-04
**Brief Summary of the Reviews**

We are thankful for the reviewers’ valuable feedback, which would help us to improve the quality of our works! Here we would like to **firstly respond to several questions** to make clarification for our proposed work:

**C1. Existing Graph Foundation Models with PCA alignment**: As we recognize in section 4, there exist works that utilize PCA to align a same dimension length for different input graphs. However, this paper is **not aiming to “claim” a new alignment method** but to **give more insights and findings** regarding the powerfulness of PCA alignment.

**C2. Latent shift between datasets:** We admit that the latent space of different graph’s features may have great shift if two datasets are too dissimilar, however, such shifts could then be **overcome by increasing the training datasets’ diversity** and the model’s **hidden width**. As our model **scales up with more training datasets** representing various latent space, the latent invariance could help to generalize to datasets which are **unseen but likely to hold similar latent space** with some of our training datasets.

**C3. Novelty of PCA on Permutation Invariance:** The motivation of our paper is to using PCA to deal a frontier problem that raised in several recent popular GFM works. While the concept of PCA is well-known and easy, we **haven’t seen discussion of its value on dimension invariance explicitly** and especially such discussion on **GFM**. Therefore we believe it’s still important to notify its advantages as a **“simple yet effective” solution to this frontier problem**.

**C4. Efficiency Concern:** While our alignment method may increase the feature dimension in some constant level, **it reduces repeating calculation for node embedding in training epochs**, which would be much higher. Besides as our experiment show its effectiveness on large graph datasets, it would be a good choice when users hope to find efficient solutions to analyze a large graph.

**C5. Cases of XS**: We didn’t study the alignment for XS but on SX due to the reason that **XS stands for the volume differences** between **different nodes’ features**. It’s possible in real-world that some **outstanding nodes’ may have much higher volume in features** compared with some others, and **such differences seem not good to be mitigated**.

**C6. Typos on $P_{X}X$**: Sorry for the typos. We found that before equation(16) in appendix we mistakenly state X’=P_{x}X, while in the later proof we still use X’=XP_{x} correctly. Thanks for raising this and **we will fix this**.

Despite the questions and suggestions on our weaknesses, we are also **thankful to find recognition of the following contributions** of our work from the reviewers:

**S1.** It’s an **important problem** to study the GFM’s generalization abilities in theoretical perspectives.

**S2.** Our proposed framework of (1) prealigning node features of pretrained graph datasets, (2) pretraining GFM on datasets from different domains and (3) generalizing to new datasets with only downstream layers training, is **simple yet efficient and novel**;

**S3.** Our **theoretical proofs** of PCA alignment on GFM are **solid and give theoretical guarantees** on both feature permutation invariance and latent space invariance, which are **valuable and meaningful**.

These recognitions inspire us to further improve our paper and present for a better version!

After studying reviewers’ opinions we also found **most of our weaknesses could be addressed by additional experiments**, including more ablation study, comparison with other baselines and testing on more kinds of datasets and downstream tasks. Our futural improvement plan mainly contains:

**P1.** We would test our mini-GFM under **more downstream tasks** including link prediction, using **more GNN architectures as the core model** and testing against zero-shot learning on **more and larger datasets**.

**P2.** To better illustrate the effectiveness of PCA for GFM and have full respect to **existing GFMs or few-shot works with PCA alignment**, we will **conduct comparison experiments on them** to test their performance **with and without PCA**, to better illustrate PCA’s importance rather than just unification of the feature dimensions.

**P3.** We will also do **more ablation study and visualization results** for better presentation and more interesting findings of PCA’s effectiveness.

**P4.** We will **collect more related GFM methods** that didn’t aim to solve permutation invariance, and **compare with them in performance** as well.

**Thanks again for our reviewers’ precious opinions and suggestions. We will work on addressing them and present our work in a better version in the future!**

---

### Meta-Review · Area_Chair_X9wU · 2026-01-09

**Summary:**

This paper focuses on addressing feature heterogeneity across graph datasets in the context of Graph Foundation Models (GFMs), proposing the Mini-GFM framework, which uses principal component analysis (PCA) to align node features across domains. The problem is timely and relevant, and reviewers acknowledge the importance of studying cross-dataset generalization and feature alignment in GFMs.

However, reviewers raise substantial concerns regarding the underlying assumptions of the method, the novelty of using PCA for feature alignment, and the rigor of the evaluations. Reviewers gzW1 and dqxB argue that the core idea of using PCA for feature alignment is not novel and has already appeared in multiple prior works, and note that the paper largely restates well-known properties of PCA rather than introducing new theoretical insights. Multiple reviewers agree that more thorough justification and demonstration of when their assumptions work in practice are needed.

The experimental evaluation is also widely viewed as insufficient. Reviewers gzW1, kSxk, and dqxB note that the evaluations are on only a small number of datasets and do not include error bars or many relevant state-of-the-art methods such as MDGFM. Reviewer kSxk also brings up an important concern that the model needs linear layer retraining for new datasets despite claims that the method is zero-shot.

Unfortunately, the authors did not provide point-by-point rebuttal responses or substantively engage with these critiques, leaving key limitations raised by the reviewers unresolved. Thus, based upon these unresolved concerns, the lack of novelty or compelling evaluations, I believe that the paper is not yet ready for publication.

**Reviewer Concerns:**

GFMs involve datasets from diverse domains and thus the use of PCA alone is likely to be insufficient. The work needs further justification for PCA alignment and a more thorough investigation into which domains/graphs the method can be used for.

**Reviewer Scores:**

The reviewers were unlikely to have changed their scores.

---

### Decision · Program_Chairs · 2026-01-26

Reject